# Different Pathways Conferring Integrase Strand-Transfer Inhibitors Resistance

**DOI:** 10.3390/v14122591

**Published:** 2022-11-22

**Authors:** Clémence Richetta, Nhat Quang Tu, Olivier Delelis

**Affiliations:** Laboratoire de Biologie et Pharmacologie Appliquée, ENS-Paris-Saclay, CNRS UMR 8113, Université Paris-Saclay, 91190 Gif-sur-Yvette, France

**Keywords:** HIV-1, strand-transfer inhibitors, integrase, unintegrated viral DNA, resistance

## Abstract

Integrase Strand Transfer Inhibitors (INSTIs) are currently used as the most effective therapy in the treatment of human immunodeficiency virus (HIV) infections. Raltegravir (RAL) and Elvitegravir (EVG), the first generation of INSTIs used successfully in clinical treatment, are susceptible to the emergence of viral resistance and have a high rate of cross-resistance. To counteract these resistant mutants, second-generation INSTI drugs have been developed: Dolutegravir (DTG), Cabotegravir (CAB), and Bictegravir (BIC). However, HIV is also able to develop resistance mechanisms against the second-generation of INSTIs. This review describes the mode of action of INSTIs and then summarizes and evaluates some typical resistance mutations, such as substitution and insertion mutations. The role of unintegrated viral DNA is also discussed as a new pathway involved in conferring resistance to INSTIs. This allows us to have a more detailed understanding of HIV resistance to these inhibitors, which may contribute to the development of new INSTIs in the future.

## 1. Introduction

The World Health Organization estimated that, since the beginning of the HIV pandemic, more than 36.3 million of people have died of AIDS. 37.7 million people were living with HIV in 2020 and 1.5 million people became newly infected with HIV in 2020. Even if there is still no treatment able to definitively cure the infection, many advances in antiretroviral drugs have been made in the search for new ways to inhibit HIV replication and to maintain an undetactable viral load.

HIV replication is composed of many steps from the viral entry into the target cells to the release of newly infectious viral particles. The reverse transcription (RT) step is the first to occur; it converts the viral RNA into a linear viral DNA molecule [1] flanked by two identical structures called LTR (Long Terminal Repeat) [2]. These repeated structures are critical in the subsequent reactions [3]. During RT, the frequency of mutations could be high due to the lack of a 3′ to 5′ exonuclease proofreading activity of the reverse transcriptase. As a consequence, the linear double stranded viral DNA generated during RT and used as the substrate for further reactions shows a high heterogeneity [4].

The linear viral DNA is then associated in a nucleo-protein complex composed by viral and cellular proteins called the pre-integration complex (PIC) [5]. This complex is then translocated into the nucleus of the target cell and the viral genome is integrated into the host cell genome during the integration step mediated by the viral protein integrase (IN) [3]. Two main reactions occur during integration of the viral DNA (Figure 1). First, the 3′ processing reaction (3′ P) that consists in an endonucleolytic reaction at the viral DNA extremities (Figure 1) [6]. During 3′P, a dinucleotide is cleaved from each end of the LTR leading to CA_OH_ reactive extremities [7]. This reaction occurs to ensure the good positioning of the viral DNA ends in the catalytic site of IN for the next step: the strand-transfer reaction (ST) (Figure 1). The resulting 3′-proccessed DNA is then used as a substrate to be inserted in the genome of the infected cell [6].

Two other activities can be also described for IN. The first one, named disintegration, is considered to be the opposite of the strand-transfer reaction. However, the disintegration reaction only requires the catalytic core to be realised while strand-transfer requires the full-length protein [8,9]. To date, no experimental data suggest that this reaction could occur in the cell. The second one has been described recently as an endonucleolytic activity on a specific sequence in vitro [10]. This activity has also been reported for the Prototype Foamy virus (PFV) IN [11]. Importantly, this internal cleavage is specific of the cognate sequence of the LTR-LTR junction found into 2-LTR circles, a circular genome found in the nucleus of the infected cell when the integration reaction is impaired. Interestingly, this activity has been revealed during viral infection [12].

It is important to note that these reactions involve a precise oligomeric state of the complex composed by IN and viral DNA. Indeed, in the case of HIV-1, a tetramer of IN is required that is stabilized by interaction with the two viral DNA extremities [13,14,15]. Determination of the precise structure of the IN/DNA complex is crucial to understand the way to inhibit the IN activity and then to develop inhibitors.

The covalently inserted genome is then called provirus and is responsible for the synthesis of new infectious viral particles. This integration step is fundamental for the retroviral replication and a common step in the retroviral family for two main reasons. First, once integrated in the host cell genome, the genetic viral information is stable and persists in the infected cell until its death. Second, the provirus is the main template for an efficient viral transcription ensuring the synthesis of new infectious viral particles.

Due to its importance in the retroviral life cycle, many efforts have been realized to prevent viral integration and thus inhibit integrase. Many compounds have been tested for their efficiency to prevent 3′P and/or strand transfer reaction. To date, the most efficient compounds, used clinically to treat patients, belong to the integrase strand-transfer inhibitors family (INSTI). INSTIs play an increasingly central role in HIV treatment thanks to their high potency for HIV inhibition and their low toxicity. HIV treatment guidelines all recommend to include INSTI-based regimens as first-line therapy [16]. However, treatment fails in some patients due to the development of resistance mechanisms by HIV-1.

Moreover, it is important to note that integrase inhibitors do not lead to the clearance of the viral DNA. Indeed, the HIV genome also exists as unintegrated forms which accumulate in infected cells under INSTI treatments [17]. Two types of unintegrated molecules can be generated by two different mechanisms. Homologous recombination of the linear viral DNA leads to the formation of 1-LTR circles whereas 2-LTR circles can be created by a non-homologous end joining mechanism [18]. Accumulation of these circular genomes under INSTI treatments could play an important role and could be involved in resistance pathways.

This review describes the mode of action of integrase inhibitors and is then focused on ways used by the HIV-1 virus to counteract the effect of INSTIs leading to resistance. Both the role of resistance mutations and the involvement of unintegrated viral DNA will be discussed.

## 2. Integrase Function Involved in Viral Integration

All members of the Retroviral family involve IN as the main viral protein responsible for the viral genome integration [19]. It is important to note that IN is also involved in other steps of the retroviral life cycle such as maturation of the viral particle and reverse transcription. (For a review, see [20]).

Maturation of the Gag-Pol precursor by the viral protease leads to the production of HIV-1 IN. This viral protein (288 amino acids) highlights 3 domains, highly conserved in the Retroviridae family. The N-Terminal domain (N-Ter, amino acids 1–49) favors the multimerization of the protein [21,22]. The Central-Domain (CC), also called the Catalytic Domain (amino-acids 50–212) has the peculiarity to contain the Catalytic triad composed by the D,D-35,E motif. This motif, highly conserved in integrases and transposases, coordinates the metallic ion (Mn^2+^ or mainly Mg^2+^) and is then responsible for the catalytic activity of the protein [23,24]. Finally, the C-Terminal domain (from amino acids 213–288) is mainly involved in the stability of the complex with DNA due to its non-specific binding to DNA [25].

Importantly, the optimal IN activity is linked to its oligomeric state [26]. Indeed, the 3′P and strand-transfer reactions can be studied in vitro using double stranded oligonucleotides mimicking the LTR end [27,28]. A consensus of many studies reports that a dimer of IN is responsible for HIV-1 3′ processing and that a tetramer is responsible for the concerted integration, i.e., integration of both extremities at the same location [14,29,30,31,32] as well as the palindromic activity on the 2-LTR junction [12].

All IN activities can be studied in vitro. Importantly, these activities are modulated by different viral and cellular proteins. The most important partner of IN is LEDGF/p75, a cellular protein that has been shown to stimulate IN activity in vitro as well as targeting the viral integration in vivo [13,33].

Importantly, IN activity can also be modulated by both viral and host DNA structures. As an example, Wang and colleagues report that alterations in the minor groove of viral DNA result in a decrease in 3′ processing in a more important manner than major groove substitutions [34]. Furthermore, concerning the host DNA, Taganov and colleagues report that the efficiency of in vitro integration was decreased after compaction with histone H1 [35]. These studies clearly demonstrate that efficiency of integration by IN is greatly influenced by the curvature and flexibility/rigidity of DNA and then influences the way by which inhibitors prevent integration.

## 3. Inhibitors

As described previously, IN catalyses several steps at a single active site (3′ Processing and Strand-transfer reaction). The fact that these two reactions are catalyzed in the same location requires a strong binding of the DNA product of the first reaction (3′ Processing) that is used as a substrate for the second step (Strand-transfer). This strong binding decreases the efficiency of the reaction in terms of turnover. However, this is not detrimental for the overall process since, in vivo, only one event of integration is necessary to maintain the viral information into the cell. However, researchers have taken into account the strong association of the 3′-processed DNA with IN to develop a strategy in order to inhibit integration.

Several families of compounds have been developed such as IN DNA-binding inhibitors (INBI) and IN Strand-transfer inhibitors (INSTI) [36,37]. INBI were developed in order to prevent the fixation of IN on DNA. However, due to their inefficiency in disrupting the pre-formed complex between IN and DNA, their development was stopped.

INSTI were successfully developed from diketo acids (DKA) inhibitors by Merck and Shionogi scientists. Some compounds, such as L-731,988, were found to inhibit the strand-transfer activity within the pre-formed complex. The property of the DKA members is their fixation into the catalytic site of integrase when viral DNA is bound [37,38]. By their fixation into the IN/viral DNA complex, close to the 3′ end of the processed viral DNA, they compete with the fixation of the host DNA. Therefore, INSTI compounds inhibit specifically the strand-transfer reaction and not the 3′ processing step [39].

The INSTI structure is composed of a metal-chelating core (in most of cases, a coplanar triad of oxygen atoms) that is optimized to bind a pair of Mg^2+^ ions in the active site and a halobenzyl side chain, connected to the core by a flexible linker (Figure 2 and Figure 3) [40]. Several molecules have been developed and used in antiretroviral therapy: Raltegravir (RAL), Elvitegravir (EVG), Dolutegravir (DTG), Cabotegravir (CAB) and Bictegravir (BIC).

### 3.1. Raltegravir

The first inhibitor belonging to the INSTI family, Raltegravir (RAL) (Isentress@), was approved to treat patients in 2007. Efficiency of the compound is high since the IC_50_ was in the nanomolar range in vitro and in vivo [41]. In some limited treatment conditions, the contribution of RAL and optimized background therapy reduces the duration of treatment at least 48 weeks, compared to the optimized background therapy alone [42]. Moreover, RAL decreases the plasma HIV-1 RNA level 100 times after only 24 weeks of treatment (from 5000 copies/mL in 85–95% of treatment-naive patients to less than 50 copies/mL) [43]. Aside from that, it should be noticed that although RAL is quite susceptible to drug resistance, it is still a potential treatment option with many advantages over other antiretroviral drugs. The STARTMRK trial (phase III non-inferiority trial of raltegravir-based versus efavirenz-based therapy in treatment-naive patients) by Rockstroh et al. in 2013 [44] showed that RAL has better efficacy and fewer side effects than efavirenz in clinical treatment. An important point is the half-life of the compound on the IN/DNA complex. Concerning RAL, Hightower and colleagues estimated its half-life at 9 h (Table 1), stronger than the one found for Elvitegravir (EVG), another strand-transfer inhibitor belonging to the first generation of anti-integrase inhibitors [45].

### 3.2. Elvitegravir

EVG was initially developed by the Central Pharmaceutical Research Institute of Japan Tobacco, Inc., (Osaka, Japan) and followed Gilead Sciences (Foster City, CA, USA). The outstanding advantage of EVG over RAL is that EVG is prepared in a combined form for the purpose of once-daily dosing. The first product under the brand named STRIBILD^®^ associates EVG at 150 mg, Cobicistat (COBI) at 150 mg, Emtricitabine (FTC) at 200 mg and Tenofovir disoproxil (TDF) at 300 mg [EVG/c/FTC/TDF] and is considered as the first INI-based single-tablet regimen administered once-daily [47]. Then, Genvoya^®^, made up with combination of EVG 150 mg/COBI 150 mg/Tenofovir alafenamide (TAF) 10 mg/FTC 200 mg [EVG/c/TAF/FTC], significantly reduced side effects on kidneys and bones by the lower-dosed single-tablet regimen, as compared to STRIBILD^®^ [47]. In this way, Genvoya^®^ has significantly improved patient adherence, which plays an important part in cases of long-term treatment such as HIV treatment.

It can be seen that the first generation of INSTIs (RAL and EVG) are both potent and highly selective IN inhibitors; EVG is even more effective than RAL. They differ in their ways of inhibiting resistant viruses. For example, HIV mutants harboring the Q148K and T66I mutations in the IN showed high resistance to both drugs, while those harboring the S153Y mutation showed higher resistance to EVG than RAL [48]. Finally, numerous reports suggest cross-resistance between RAL and EVG [49]. This eliminated the possibility of using EVG-based therapies when RAL-based treatment fails. Therefore, there is an urgent mission to introduce a second generation of INSTIs, which solves problems such as drug resistance or cross-resistance.

### 3.3. Dolutegravir

In 2013, Shionogi and GlaxoSmithKline developed a carbamoyl pyridone HIV-1 integrase inhibitor as a second generation INSTI, which overcame some severe disadvantages of the previous generation, including limited or no-cross resistance to early generation INIs [50]. Although quite similar to RAL and EVG in term of structure, the peculiarities of DTG (length and flexibility of the linker bridging its tricyclic metal-chelating core and the difluorophenyl ring) is responsible for the higher potency of inhibition of DTG compared to the previous compounds, see below (Figure 3).

According to VIKING Study of Joseph J Eron in 2012 [51], the in vitro resistance profile of DTG (compared to RAL and EVG) demonstrated activity against site-directed molecular clones containing the raltegravir-resistant signature mutations Y143R, Q148K, N155H and G140S/Q148H. DTG has demonstrated outstanding efficacy against HIV-1 replication at low nanomolar or subnanomolar potency with notable figures such as: its EC50 against HIV-1 was 0.51 nM in PBMCs, 0.71 nM in MT-4 cells and 2.2 nM in a PHIV assay. Its mechanism of action as an INSTI was further demonstrated by a variety of in vitro assays including its potent inhibition of integrase strand transfer activity with an IC_50_ of 2.7 nM [45]. Because of these significant effects, DTG is approved for use in HIV-infected patients (treatment-naive or treatment-experienced, including those already treated with other INSTIs) and also for children ages 12 years and older, weighing at least 40 kg [52].

**Figure 3 viruses-14-02591-f003:**
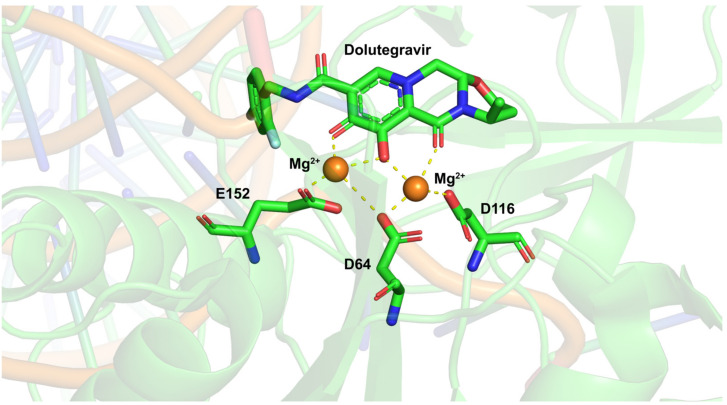
Mode of action of dolutegravir targeting HIV-1 integrase. This figure shows the 3D structure of the HIV-1 IN catalytic site complexed with DTG. The three coplanar oxygen atoms (in red) of DTG chelate Mg^2+^ cations (in orange) in the intasome active site constituted by the D116, D64 and E152 residues. The length and flexibility of the linker bridging the tricyclic metal-chelating core and the difluorophenyl ring are responsible for the high potency of inhibition of DTG. This 3-D structure was prepared with The PyMOL Molecular Graphics System, Version 1.3, Schrödinger, LLC using the PDB (protein data bank) 6VDK [53]. Red: oxygen atoms, blue: nitrogen atoms.

The half-life (T_1/2_) of the INSTI on the IN/DNA complex is an important parameter. Indeed, a high T_1/2_ of the compound leads to a higher inhibition of the complex over time. The half-life, in theory, represents the time it takes for a drug to bind to its target, which is directly proportional to the drug’s ability to exert its effects.

The half-life of RAL on the IN/DNA complex is weaker than the one found for the INSTI belonging to the second generation of inhibitor such as DTG [45]. Indeed, the dissociative half-life of DTG on the pre-formed complex is 10 fold stronger than the one calculated for RAL (71 h and 9 h for DTG and RAL, respectively) (Table 1) [45]. Moreover, when observing the effect of DTG on mutant strains, T_1/2_ of this compound also showed a better result than the first generation: 42 h for Y143R, 11 h for Q148K and 9.6 h for N155H, whereas T_1/2_ values for RAL are significant lower (2.5 h, 0.3 h and 0.6 h, respectively) [45]. The long half-life of DTG on PIC (pre-integrase complex) highlights its ability to fight HIV-1 (WT) and also the first generation INSTI-resistant mutants.

Currently, DTG is recommended for its use in first-line combination antiretroviral therapy but not in monotherapy assay (WHO Policy Brief 1–16).

### 3.4. Bictegravir and Cabotegravir

Bictegravir (BIC) and Cabotegravir (CAB) are the two most recent inhibitors, developed by ViiV Healthcare and Gilead Sciences, respectively [54,55].

One characteristic of Bictegravir is that BIC highlights a unique bridged bi-cyclic ring and a distinct benzyl tail (Figure 2). BIC has displayed improved resistance compared with RAL, EVG and DTG, particularly for high level INSTI resistance containing combinations of mutations such as E92Q+N155H or G140C/S+Q148R/H/K [56]. Importantly, BIC show an interesting antiviral synergic effect when it is associated with other compounds such as emtricitabine and tenofovir alafenamide as well as with darunavir [56]. Biktarvy^®^, a single tablet anti-HIV medication combining BIC with FTC and TAF developed by Gilead Sciences, was approved by the FDA for use in the United States and globally in February 2018 for treatment-naive patients or to replace current ART regimens in individuals who have achieved viral suppression (HIV-1 RNA/50 copies/mL) for ≥3 months.

Cabotegravir highlights a structure similar to DTG (Figure 2) by following the carbamoyl pyridine class [57] and is a very promising drug since it can be used as a long-acting drug (LA-CAB). Results of the phase I study show that LA-CAB remains efficient in preventing HIV infection after 16 weeks from the first exposure [58]. CAB has many outstanding advantages such as good pharmacokinetic characteristics, long half-life allowing for low-dose and once-daily dosing [59]. It does not require to be co-administered with CYP450 inhibitors such as DTG [60] and was shown to be well tolerated at all dose levels with no clinical evidence of resistance [61].

Although INSTIs are powerful antiretroviral molecules, HIV-1 is able to counteract their actions by developing resistance mechanisms. Mutations in the viral genome and in particular in the IN gene play a major part in resistance pathways.

## 4. Mutations Leading to Resistance

### 4.1. Substitution Mutation

Substitution mutations in the IN gene are the first type of mutations of the HIV genome which significantly reduced the therapeutic effect of the first generation of INSTIs, RAL and EVG. In many cases, these primary mutations not only confer resistance to INSTIs but also reduce the virus’s ability to replicate.

Three major IN signature mutation pathways result in significant loss of potency of RAL (N155H, Y143C and Q148H(R)(K)) and render it ineffective against these mutant viruses [62,63]. These pathways are similar for both HIV-1 and HIV-2 IN [64]. EVG treatment engenders slightly different resistance pathways (T66I, E92Q, S147G, Q148H(R)(K), N155H). Taking an example, the Y143C substitution mutation leads to RAL resistance but not EVG resistance [65]. However, in most of the remaining well-known substitution mutants, notably N155H and G140S/Q148H, neither RAL nor EVG could strongly inhibit these pathways. The rationale for this growing resistance may come from their similarities: these two INSTIs bind to the active site of the intasome.

For the second generation of INSTIs, typically DTG, the addition of a hydroxyl group to the piperazinone series has yielded a desirable result. Resistances due to substitution mutations that cause failure in first-generation INSTIs (Y143R, N155H and G140S/148H for RAL, as well as T166I and E92Q for EVG) have been impaired [57]. When compared with resistance to first-generation INSTIs, resistance to DTG has been much more modest [66]. Although DTG resolved the main resistance pathways that emerged during treatment with RAL and EVG, some mutations responsible for resistance to second-generation of INSTIs, such as DTG, are still selected and are responsible for treatment failure. One example commonly selected by DTG was R263K [67]. Additional mutants have been reported to be selected by DTG, including H51Y, T66A/I, G118R, E138K, and S153Y/F [68].

### 4.2. Insertional Mutation

Unlike previous widely reported substitution mutations, a new type of mutation, never described until now, has been recently reported as a mechanism of resistance. These mutations consist of the insertion of several amino acids in the IN and have recently appeared in HIV-2 (Figure 4). Even if insertional mutations have been poorly described, they cause quite a lot of serious consequences, typically being resistant to second generation INSTIs.

In 2019, Quentin Le Hingrat et al. reported 5-amino-acid (AA) insertion at codon 231 in the integrase C-terminal domain (231INS) of 4 virus isolates coming from 3 different INSTI-experienced HIV-2 infected patients at the time of virological failure (VF) of an INSTI-based regimen. They determined the susceptibility of these isolates to the 5 different INSTIs currently marketed or under development: RAL, EVG, DTG, CAB and BIC [69].

All 4 isolates with 231INS showed a steeply reduced susceptibility to the first generation INSTIs (from 56- to 150-fold for RAL and from 8- to 149- fold for EVG). They also exhibited a slightly reduced susceptibility to the second generation INSTIs (from 3.3- to 13-fold for DTG and from 5- to 79-fold for CAB). Two isolates remained susceptible to BIC (IC_50_ fold changes of 0.2 and 0.6) whereas the other 2 isolates, obtained from the same patient, the first at virological failure of a first RAL-based regimen and the second at VF of a DTG-based regimen 4 years later, showed a slight increase in IC_50_ to BIC (5.5- and 2.9-fold change, respectively). To complete this phenotypic analysis, Le Hingrat et al. analysed the integrase sequence of 99 viruses from patients failing an INSTI-based regimen (database from ANRS HIV-2 cohort). They identified 6 viruses with insertions in codon 231. Interestingly, the sequences of these insertions were highly conserved between the 6 viruses. For 2 viruses, the insertion consists of a duplication of 15 nucleotides (nucleotides of codons 227–231). For 2 others viruses, the insertion also corresponds to a duplication of the 15 nucleotides but with one and 2 nucleotides changes. For the 2 remaining viruses, the duplication pattern was less marked since only the last 6 nucleotides of the insert were similar to those at codons 230 and 231 [69].

Pham et al. made a statement in 2019 that HIV-1 is less likely to become resistant via insertional mutations due to structural differences [70]. However, in 2021 another study identified a new 2-amino acid insertion in the integrase coding region of HIV-1 subtype G isolates [71]. Although this mutation has not yet been reported for drug resistance in HIV-1 strains, it has opened up a view on the problem of insertional mutations in both HIV-1 and HIV-2.

In conclusion, several kinds of mutations (substitutions or insertions) in HIV IN are involved in resistance pathways. However, in some patients failing INSTI-treatments in particular DTG, no mutation in the integrase gene has been reported [72]. This suggests that HIV is able to resist INSTI with a WT integrase, using new resistance pathways that could involve the use of unintegrated viral DNA.

## 5. Role of Unintegrated Viral DNA in Resistance to INSTIs

As mentioned previously, inhibition of the integration step by an INSTI blocks viral replication but does not suppress HIV viral DNA that can persist as unintegrated DNA (uDNA). Several forms of unintegrated DNA can be found. The most abundant form is the linear DNA issued from the reverse-transcription step, which is the substrate for the integration reaction [73]. The other forms of uDNA are circular forms harboring one or two LTR called 1-LTR circle (1-LTRc) and 2-LTR circle (2-LTRc) respectively. 1-LTRc and 2-LTRc are mainly found in the nucleus of infected cells. While the amount of 2-LTR is quite low (2–5% of total viral DNA in infected cells), the amount of 1-LTRc can reach 20–30% of the viral genome [74]. Inhibition of viral integration by INSTI or using a catalytic mutant of IN, leads to accumulation of uDNA and more particularly of circular viral DNA [73,74,75]. Indeed, the amount of 1-LTRc can reach 50% of total viral DNA. The greatest accumulation is observed for 2-LTRc that can be increased by a 10-fold factor [74]. Interestingly, gene editing by CRISPR/Cas9 to excise HIV-1 provirus (cleavage in the LTR) increases proviral DNA circles meaning that the excised DNA can form DNA circles with restored LTR [76].

Several mechanisms are involved in the formation of circular HIV DNAs. Most of the uDNA are derived from the linear DNA generated by RT. Autointegration can lead to the formation of truncated or internally rearranged circular forms [77]. It is now well established that 2-LTRc are the products of the non-homologous end-joining (NHEJ) DNA repair pathway induced as a protective host response to the presence of double stranded DNA [18]. The cellular factors Ku80, XRCC4, and ligase 4 involved in the NHEJ pathway have been shown to be required for the formation of 2-LTRc [78,79]. The formation of 1-LTRc is mainly due to homologous recombination of linear DNAs at the LTRs, resulting in a circular DNA bearing one copy of the viral LTR [18]. The MRN complex (MRE11/RAD50/NBS1) is involved in this process [18]. However, 1-LTR circles can also be formed by ligation of interrupted reverse transcription intermediates [80]. A significant proportion of 1-LTRc can be generated in the cytoplasm of infected cells during RT [74].

Even if some studies have reported that 2-LTRc are short-lived molecules [75,81,82], the circular forms of uDNA are quite stable compared to the linear DNA which is rapidly degraded. Several studies show that the decrease of circular DNA within infected cells is only due to the cell division [74,83,84]. Therefore, uDNA can persist a long time in cells with a weak division rate. For example, 2-LTRc can persist for up to 21 days post-infection in macrophages [85]. It has also been shown that 2-LTRs are stable in naïve primary T CD4+ cells over a 30 day culture [86]. Moreover, integrase lentiviral vectors used for gene therapy in animal models show a very high stability in non-dividing cells for extended periods [87,88]. Therefore, this form of uDNA has the capacity to persist in slow or non-dividing cells. uDNAs have been considered for a long time as a by product of reverse transcription with no significant role in HIV replication [89]. However, their accumulation in cells treated with INSTI combined with their persistence could play an important role in resistance pathways.

### 5.1. The 2-LTR Circles, a Reservoir of HIV-1 Genomes

As we discussed, the 2-LTRc are the uDNA which are the most increased when the catalytic activities of the integrase are inhibited [74]. A peculiar feature shared by many retroviruses is the presence of a palindromic sequence at the LTR-LTR junction. Several studies have shown that recombinant HIV-1 integrase can cleave in vitro both oligonucleotides and circular DNA containing the palindromic junctions leading to the formation of linear 3′-processed–like DNA [10,12,90]. The palindrome cleavage occurs via a two-step mechanism leading to a blunt-ended DNA product, followed by a classical 3′-processing reaction [91]. This linear DNA can then be integrated into the target DNA via an IN-dependent mechanism leading to the 5-bp duplication of the host genome [91]. Thus, 2-LTRc which accumulate in cells treated with an INSTI may constitute a reserve supply of HIV-1 genomes for proviral integration. Therefore, if the block of an INSTI is removed, accumulated 2-LTR could be integrated de novo and revive viral replication (Figure 5). In accordance with this observation, it has been shown that Raltegravir inhibition is reversible and that 2-LTRc are involved in the resumption of viral integration [12]. The removal of RAL in cells leads to a decrease in the 2-LTRc amount leading to a linear intermediate that is subsequently followed by new integration events [12]. These results show that 2-LTRc can rescue viral replication after their integration in the host cell genome. Thus, they could play a crucial role for viral resurgence in the case of interruption of effective INSTI treatment.

### 5.2. Expression of uDNA

The main question about uDNA is their capacity to be expressed and to lead to the production of new infectious viral particles. All reports agree that uDNA expression is weaker compared to integrated viral DNA. Indeed, HIV uDNA are rapidly loaded with histones H3 and H2B which are marked by epigenetic modifications associated to silencing such as huge H3K9me3, and low H3 acetylation [92]. However, studies using integrase-defective viruses show that uDNA can be transcribed [93,94,95,96,97,98]. uDNA transcription generates all classes of multi-spliced, singly spliced and unspliced viral mRNAs but not in the same proportions that those observed during transcription of integrated DNA. Indeed, multi-spliced RNAs encoding viral early genes such as *nef* and *tat* are abundant while singly spliced and unspliced RNAs are poorly transcribed [95,96,97,98]. The low amount of Rev protein, which is necessary for the nuclear export of single spliced and unspliced RNAs encoding late viral genes, could explain the weaker viral replication from uDNA compared to integrated DNA. In accordance with this, providing Rev *in trans* can rescue late gene synthesis [95].

Due to the weak stability of linear DNA in infected cells, we can assume that expression of uDNA is mainly due to circular genomes. Transfection of HeLa cells with molecular constructs mimicking 1-LTRc or 2-LTRc show that both kinds of circular genomes lead to p24 production [99]. Even if a specific type of mRNA transcribed form 2-LTRc has been detected [100], Cara et al. demonstrated that uDNA expression from 1-LTRc is stronger than from 2-LTRc [99]. In any case, uDNA expression is an order of magnitude less than integrated DNA expression [99]. This finding combined with the fact that 1-LTRc are more abundant in cells compared to 2-LTRc, suggest that 1-LTRc could have a major contribution in transcription from uDNA (Figure 5).

The detection of viral transcripts from uDNA does not ensure the synthesis of viral proteins and the production of new infectious viral particles. Only the accessory and regulatory proteins Nef [96,101], Tat [93,102] and Rev [94,103] were found to be translated in detectable amounts. However, several studies clearly demonstrate that, under specific conditions, HIV-1 replication could occur without integration [104,105]. Viral particles production from uDNA has been shown in resting CD4+ after their reactivation [106].

### 5.3. Mutations in the 3′-PPT: A Way to Escape from INSTIs Using uDNA

As we described in Section 4, several kinds of mutations in the integrase gene can lead to resistance to INSTIs. However, in some patients treated with DTG, treatment failure is not associated with mutation in the integrase gene [72]. This suggests that HIV is able to resist INSTIs with a WT integrase, using new resistance pathways based on mutations located outside the target gene [107]. For example, it has been shown that mutations in the HIV Long Terminal Repeat can confer resistance to INSTIs [108]. In 2017, our team has reported the in vitro selection of a virus highly resistant to INSTIs with mutations in the 3′-PPT (polypurine tract) and not in the target gene [109]. The 3′-PPT is located at the PPT-U3 junction and consists of 15 nucleotides from positions 9059 to 9073 (5′-AAAAGAAAAGGGGGG-3′). It is highly conserved in most retroviruses and used as the site of plus-strand initiation involved in viral reverse transcription [110]. The selected virus has several mutations: a replacement of the cytidine by a thymine in position 9053 and a change of the “GGGGGG” sequence by the “GCAGT” sequence from position 9068 to 9073 including a deletion of a guanine in position 9073 [109]. Recently, we have characterized in more detail how these mutations in the 3′-PPT confer resistance to DTG [111]. First, our data clearly show that the 3′-PPT mutant is able to replicate and to produce infectious viral particles without integration. Indeed, inhibition of integration with DTG or using a catalytic mutant of the integrase (D116N) does not affect the replication of the 3′-PPT mutant [111]. Quantification of the different forms of viral genomes highlight the absence of integrated DNA associated with an accumulation of 1-LTRc but a very low amount of 2-LTRc. Importantly, we detected only a very small amount of linear viral DNA in infected cells, whether in the presence or absence of DTG, suggesting that 1-LTRc were formed directly during reverse transcription and not by homologous recombination of the linear viral DNA [111]. Accumulation of 1-LTRc with a decrease of linear viral DNA has been previously reported for a lentiviral vector with a deletion of the 3′-PPT [112]. Using biochemical experiments, we have shown that mutations in the 3′-PPT lead to the modification of the reverse transcription step. The altered sequence of the mutated 3′-PPT may compromise its efficiency as a primer for the initiation of + stranded DNA synthesis during RT. Most probably, the + strand synthesis starts from the cPPT and the length of the so formed DNA fragment impedes the occurrence of the + strand transfer needed for the formation of the complete LTRs at both ends, required for an integration-competent cDNA [111,113]. Taken together, our data demonstrate the replication of the virus, even at a low level compared to the WT, encompassing the mutation in the 3′-PPT region involving unintegrated viral DNA (1-LTR circles), explaining the resistance to INSTI (Figure 5).

Other studies about the impact of the mutations in the 3′-PPT on INSTI resistance have been made with conflicted results [114,115,116]. The work of Hachiya et al. confirms that mutations in the 3′-PPT lead to INSTI resistance [114] whereas the studies of Smith et al. [115] and Wei et al. [116] do not. These apparent discrepancies could be explained by the different settings of the experiments involved. An important difference between the different studies is the strain of HIV-1 used. In our work, the initial selection of the 3′-PPT mutant virus was made using an HIV-1 Lai strain which is highly infectious [109]. In contrast, other studies have used molecular clones of HIV-1 such as pNL101 [114] or HIV-1 vectors such as pNLNgoMIVR-Emod Luc [115] which are less or not replicative. These results could suggest that, depending on the viral replication capacity due to the viral backbone, the level of resistance to INSTI conferred by 3′-PPT mutations is not the same.

Even though DTG is a highly potent integrase inhibitor, because of its stability on the integrase/DNA complex and its high genetic barrier, our data on the 3′-PPT mutant virus could explain why mutations in the integrase gene could not be detected in patients failing DTG-based treatment [72]. Importantly, mutations in the 3′-PPT have been confirmed in vivo in a patient failing to respond to DTG without mutations in the integrase gene [117].

## 6. Conclusions

INSTI are potent inhibitors of HIV-1 replication by impairing retroviral integration, a critical step in the generation of infectious viral particles. All INSTIs used in clinics have similar potential to inhibit the WT HIV-1 virus. As described in this review, resistance to INSTI is a public health issue and is mainly due to mutations in the integrase gene (Figure 5). For RAL, several pathways of resistance are described and well characterized. This is more obscure for more advanced inhibitors such as DTG. Indeed, as mentioned previously, unintegrated viral DNA seems to play an important role in the resistance to these inhibitors. One hypothesis can be advanced to explain this observation and is based on the half-life of the inhibitor on the intasome (Table 1 and Table 2). When the half-life of the compound (for example RAL) is not sufficient to completely and definitively block the intasome before its degradation, the compound is “released” from the complex and some integration events can occur. If integration of viral DNA encompassing a mutation in the integrase gene potentially responsible for resistance (generated from mutations due to the reverse transcription step) happens, production of viral particles with the mutation can occur and leads to the propagation of the mutant. In the case of more “stable” compounds on the intasome, all IN/DNA complexes competent for viral integration are blocked and no integration can occur. Then, if mutations appear, such as mutations in the integrase gene, replication of these mutants from unintegrated viral DNA could be too weak to lead to their propagation. However, if mutations occur in a region conferring an advantage for replication using unintegrated viral DNA (for example, 3′-PPT mutation), propagation of the virus, even at a low level, will be possible and could lead to a viral escape. Studies to elucidate the precise mechanism of replication from unintegrated viral DNA of the 3′-PPT mutant are needed. Nevertheless, particular attention should be paid to the potential role of unintegrated viral DNA, more particularly in the question of viral resistance to INSTI.

## Figures and Tables

**Figure 1 viruses-14-02591-f001:**
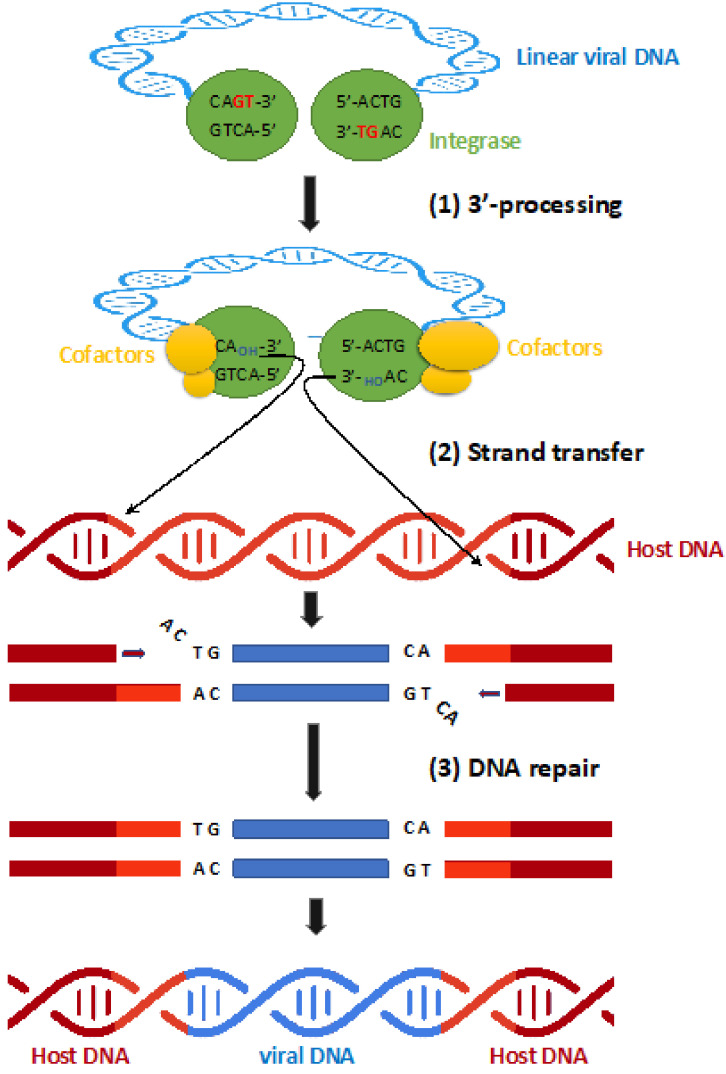
The different steps of HIV DNA integration into the host genome. (Step 1) 3’ -Processing: The integrase mediates the endonucleolytic cleavage of the two nucleotides GT to expose sticky CA-3’-OH at both 3’-ends of viral cDNA, which forms the pre-integration complex (PIC) with viral and host cofactors. (Step 2) Strand transfer: by transesterification reaction, the hydroxyl groups at the 3’-ends of the processed viral DNA are used to attack opposite strand phosphodiester bonds separated by 5 bases in the 5′ direction, which allows the integration of viral DNA (blue) into the host DNA (red). (Step 3) DNA repair mechanisms allow completion of the process yielding a duplication of host DNA flanking the provirus (light red).

**Figure 2 viruses-14-02591-f002:**
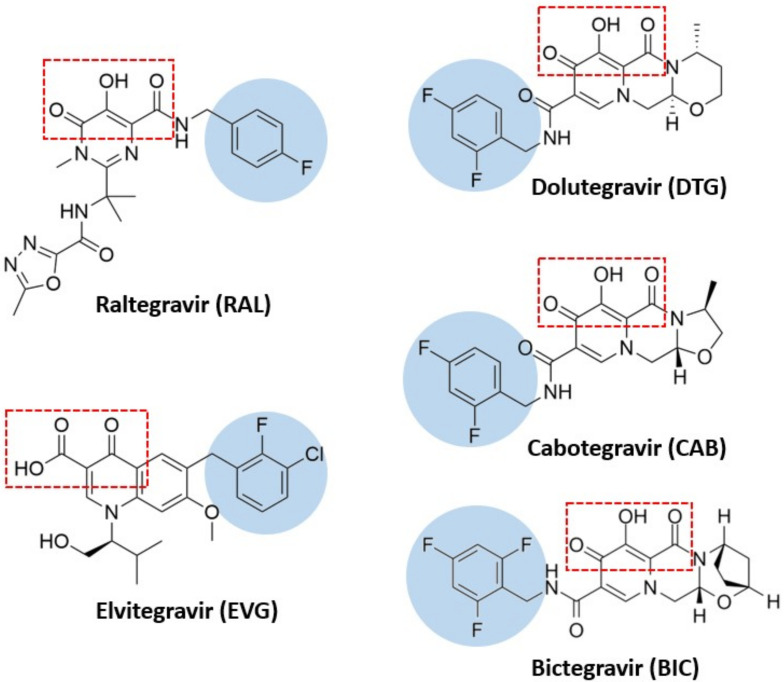
Chemical structures of the clinically relevant INSTIs. The chelating motifs which interact with Mg^2+^ cofactors in the integrase active site are highlighted in the red square. The halobenzyl moieties, which are connected to the centralized pharmacophore by a linker group, are in the blue circle. These chemical structures have been drawn using ChemDraw software.

**Figure 4 viruses-14-02591-f004:**
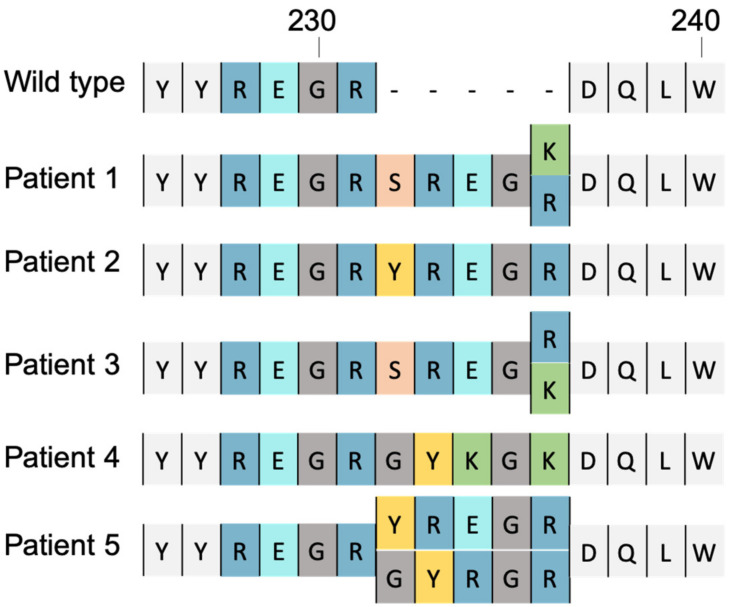
Sequences of wild-type and insert-containing human immunodeficiency virus type 2 (HIV-2) integrase (from codon 226 to 240). Alignment of amino acids sequence of the reference strain and sequences of integrase found in patients after initiating the RAL-based regimen. Insertions described by Le Hingrat and colleagues lead to a high level of RAL and EVG resistance and a moderate resistance to DTG and CAB [69].

**Figure 5 viruses-14-02591-f005:**
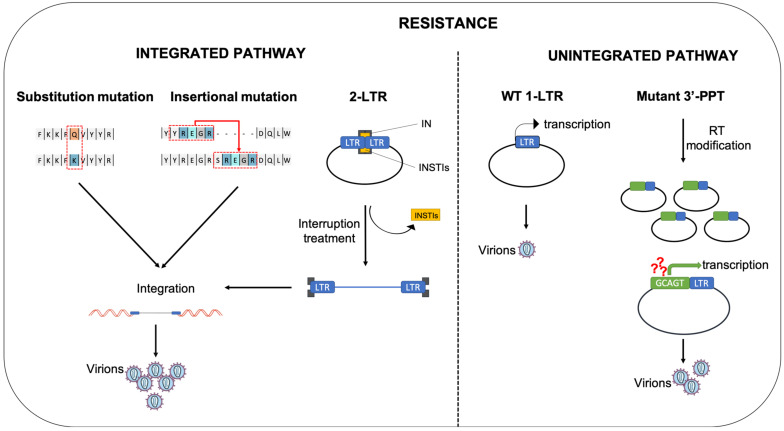
Different pathways of INSTI resistance. Two main pathways involving the integration step or unintegrated viral DNA can be responsible for INSTI resistance. Substitution mutations (replacement of one amino-acid by another one) or insertional mutations (insertion of one or several amino-acids) in the integrase protein can lead to INSTI resistance. In these pathways, integration of the viral DNA can occur because the mutated integrase is resistant to INSTIs but still active to mediate integration. Integrated DNA then allows the production of new viruses bearing these resistance mutations. Blocking of integration by INSTI treatment leads to accumulation of unintegrated DNA, in particular, 2-LTR circles. In case of interruption of treatment, integration from 2-LTR circles can occur due to the specific cleavage of the LTR-LTR junction before integration after release of the INSTI compound from the IN/DNA complex. Thus, 2-LTRc accumulated in cells treated with an INSTI may constitute a reserve supply of HIV-1 genomes that could be integrated de novo and revive viral replication. Unintegrated viral DNA could also contribute to INSTI resistance by a basal production of viruses from 1-LTR circles. Furthermore, mutations in the 3′-PPT region modify the reverse transcription step. In this case, reverse transcription does not result in linear DNA but in a 1-LTR circle. One hypothesis is that the 3′-PPT mutation leads to a higher transcription efficiency leading a higher viral production compared to the production from a 1-LTR circle that does not highlight the 3′-PPT mutation. Viruses with 3′-PPT mutations are resistant to INSTIs since their replication is independent of the integration pathway. Yellow square: INSTI. Black arrow: basal transcription from 1-LTR circles coming from a WT infection. Green arrow: hypothesis of higher transcription due to the 3′-PPT mutation.

**Table 1 viruses-14-02591-t001:** Dissociation half-life (T_1/2_) of INSTI.

INSTIs	Dissociative T_1/2_ (h)	Reference
RAL	8.8	[45]
EVG	2.7	[45]
DTG	71	[45]
BIC	163	[46]

**Table 2 viruses-14-02591-t002:** Possible correlation between INSTI dissociation half-life (T1/2) and resistance.

INSTIs	Dissociative T_1/2_ (h)	Possible Integration Events Involving uDNA (2-LTR Circles or Linear Viral DNA)	Emergence of Resistant Strains with Mutations in the IN Gene	Emergence of Resistant Strains without Mutations in the IN gene
RAL	8.8 [45]	++	+	−
EVG	2.7 [45]	++	+	−
DTG	71 [45]	+/−	+/−	+
BIC	163 [46]	+/−	+/−	+

When dissociation half-time is weak, in the case of RAL and EVG (two first lines), some integration events can occur involving uDNA when the INSTI is dissociated from the IN/DNA complex. Integrated DNA can produce infectious viral particles infecting new cells. New infections can lead to mutations in the integrase gene during the reverse-transcription step conferring resistance to INSTI. Propagation of the virus leads to the emergence of resistant strains. When the dissociation half-time is longer, as in the case of DTG or BIC (third and fourth lines), the IN/DNA complex is locked and no integration can occur. The only way to sustain the viral information is the weak replication from uDNA. As described previously, a weak amount of infectious viruses can be produced from uDNA leading to the infection of new cells. During the reverse transcription step, mutations, outside the integrase gene, can be selected conferring an advantage for the virus for its replication involving uDNA.

## Data Availability

Not applicable.

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
