# Peer review of "Different Pathways Conferring Integrase Strand-Transfer Inhibitors Resistance"

_viruses, 2022, doi:10.3390/v14122591_

Round 1

Reviewer 1 Report

Excellent and very complete review on INSTIs. Very well written. Nice figures and well explained issues on resistance with these specific category of compounds.

I would add integrase also in the title of the review. Also "and al" I would do everywhere "and al." for refering to other authors.

Reviewer 2 Report

1.    How relevant is the circular unintegrated DNA in development of drug resistance, in other words what is the reported incidence of this mechanism of resistance development in clinical trials

2.    It is helpful to the reader to include a diagram that illustrates the basic integration process of the virus as described in the text.

3.    Please insert a reference for figures 1-2 if derived from a reference, otherwise if created by authors please reference the software used to create the chemical structures.

4.    Please consider explaining figure 4 further to make it easy for the reader to interpret without reading the text

5.     “One hypothesis can be advanced to explain this observation and is based on the half-life of the inhibitor on the intasome” it is also advisable to include a diagram to explain the proposed hypothesis

Reviewer 3 Report

The manuscript of Different pathways conferring strand-transfer inhibitors resistanceby Richetta et al. is well written, and can be published in Viruses. Some minor issues:

1. There are crystal structures available for HIV-1 integrase complexed with dolutegravir or other INSTIs in the PDB, therefore, Figure 2 can be modified to a 3D figure which can be gotten from the crystal structures.

2. More figures or tables which summary the information of the manuscript are needed to make this manuscript more readable.

3. IC50 in line 163, the 50 should be superscript.
